# LC–MS/MS Phytochemical Profiling, Antioxidant Activity, and Cytotoxicity of the Ethanolic Extract of *Atriplex halimus* L. against Breast Cancer Cell Lines: Computational Studies and Experimental Validation

**DOI:** 10.3390/ph15091156

**Published:** 2022-09-16

**Authors:** Amine Elbouzidi, Hayat Ouassou, Marouane Aherkou, Loubna Kharchoufa, Nada Meskali, Abdellah Baraich, Hamza Mechchate, Mohamed Bouhrim, Abderrazak Idir, Christophe Hano, Hassan Zrouri, Mohamed Addi

**Affiliations:** 1Laboratoire d’Amélioration des Productions Agricoles, Biotechnologie et Environnement (LAPABE), Faculté des Sciences, Université Mohammed Premier, Oujda 60000, Morocco; 2Faculty of Sciences, University Mohammed First, Boulevard Mohamed VI, B.P. 717, Oujda 60000, Morocco; 3Team of Experimental Oncology and Natural Substances, Cellular and Molecular Immuno-Pharmacology, Faculty of Science and Technologies, Sultan Moulay Slimane University, Beni-Mellal 23000, Morocco; 4Faculty of Sciences, University Sidi Mohamed Ben Abdellah (USMBA), B.P. 1796, Fez 30003, Morocco; 5Laboratoire de Biologie des Ligneux et des Grandes Cultures, INRAE USC1328, University of Orleans, CEDEX 2, 45067 Orléans, France; 6Laboratory of Bioresources, Biotechnology, Ethnopharmacology and Health, Faculty of Sciences, Mohammed First University, Boulevard Mohamed VI, B.P. 717, Oujda 60000, Morocco

**Keywords:** *Atriplex halimus* L., antioxidant activity, cytotoxicity, breast cancer, computational study, ADMET analysis, toxicity prediction

## Abstract

*Atriplex halimus* L., also known as Mediterranean saltbush, and locally as “Lgtef”, an halophytic shrub, is used extensively to treat a wide variety of ailments in Morocco. The present study was undertaken to determine the antioxidant activity and cytotoxicity of the ethanolic extract of *A. halimus* leaves (AHEE). We first determined the phytochemical composition of AHEE using a liquid chromatography (LC)–tandem mass spectrometry (MS/MS) technique. The antioxidant activity was evaluated using different methods including DPPH scavenging capacity, β-carotene bleaching assay, ABTS scavenging, iron chelation, and the total antioxidant capacity assays. Cytotoxicity was investigated against human cancer breast cells lines MCF-7 and MDA-MB-231. The results showed that the components of the extract are composed of phenolic acids and flavonoids. The DPPH test showed strong scavenging capacity for the leaf extract (IC_50_ of 0.36 ± 0.05 mg/mL) in comparison to ascorbic acid (IC_50_ of 0.19 ± 0.02 mg/mL). The β-carotene test determined an IC_50_ of 2.91 ± 0.14 mg/mL. The IC_50_ values of ABTS, iron chelation, and TAC tests were 44.10 ± 2.92 TE µmol/mL, 27.40 ± 1.46 mg/mL, and 124 ± 1.27 µg AAE/mg, respectively. In vitro, the AHE extract showed significant inhibitory activity in all tested tumor cell lines, and the inhibition activity was found in a dose-dependent manner. Furthermore, computational techniques such as molecular docking and ADMET analysis were used in this work. Moreover, the physicochemical parameters related to the compounds’ pharmacokinetic indicators were evaluated, including absorption, distribution, metabolism, excretion, and toxicity prediction (Pro-Tox II).

## 1. Introduction

Cancer is a complex disease of immortal cells. Abnormal cell division and cell death are two predominant processes in the development of tumor cell invasion and tissue metastasis [1]. Additionally, cancer also occurs due to alterations in the genetic, epigenetic, and transcription factors [2]. In Africa, cancer represents between 10 and 20% of the conditions treated, and a nearly 100% increase is expected by 2030 [3]. It has been suggested that socioeconomic environment is biologically incorporated and associated with different epigenetic markers that promote the development and progression of many chronic health conditions including cancer [4]. To date, various types of cancer have been identified, among them, breast cancer (BC), the third leading cause of death in women [5].

Breast cancer is one of the most frequent malignancies and leading causes of mortality in women [6]. There were 2,261,419 new cases and 684,996 fatalities worldwide in 2020 alone [7]. Moreover, it has been reported that mortality due to cancer is about 3500/million population annually around the world [8]. Breast cancer is a very diverse disease with distinct patient tumors (intra-tumoral heterogeneity and inter-tumoral heterogeneity) [9]. Despite significant advancements in breast cancer diagnosis and therapeutic targets, breast cancer is slowly moving into the main type of chronic illnesses. Furthermore, treatment options and outcomes for breast cancer are dependent on the subtypes and involve hormonal, radiotherapy, molecular, and chemotherapy interventions [10]. However, increasing tumor heterogeneity, resistance to anti-cancer therapies (radiation therapy (RT), chemotherapy (CT), and endocrine therapy (ET)), and the high expense of existing therapeutic options are some of the current issues associated with effective management of breast cancer [11].

Cisplatin (cis-diamminedichloroplatinum II), known as CP, is an anti-cancer drug [12] that has been used for the treatment of different types of cancer (breast, testicular, ovarian, cervical, head, neck, and lung cancers) with excellent clinical outcomes [13,14]. It is generally known that CP promotes DNA damage, which causes a number of signal transduction pathways to be activated [15,16,17]. Nevertheless, it is still unclear as to exactly how cisplatin works and how particular it is. A larger section of the world’s population still uses herbal remedies and traditional doctors as their main sources of medical care. By helping cancer cells adapt and survive, which primarily results in chemoresistance, stress granules, also known as SGs (condensed non-membrane cytoplasmic granules), are formed as a response to stress exposure. This makes them a potential target for overcoming chemotherapy resistance and delaying the development of breast cancer as well [18,19,20]. Several factors can elicit SGs formation such as hypoxia, nutrient deficiency, and the important triggering factor of oxidative stress [21,22]. ROS (reactive oxygen species) perspective can be used as a novel strategy for breast cancer therapy and may be helpful as they are highly correlated to tumorigenesis [23]. It is well recognized that oxidative stress has a significant link with tumor regulation, particularly during the initiation and progression phases, allowing us to assess the relevance of this property [24]. Therefore, antioxidants may act as an efficient alternative in decreasing tumor cells in order to face drug-induced resistance. Furthermore, it has been well established that medicinal plants are believed to possess potent antioxidant capacity that subsequently contributes to the anti-cancer and anti-inflammatory abilities, especially toward breast cancers [25,26,27].

In this sense, the phytochemicals as flavonoids, polyphenols, alkaloids, saponins, terpenes, and other metabolites present in plant extracts are known to demonstrate anti-cancer properties [28,29]. In addition, the secondary metabolites from medicinal plants have been discussed in detail to possess a remarkable capacity to scavenge free radicals and inhibit lipid peroxidation, which protects the body system from oxidative damage and subsequently reduces the risk of unwanted cell mutations [30,31].

*Atriplex halimus* L. is an aromatic plant belonging to the Amaranthaceae family. It is commonly known as “Rghel” and “Lgtef” [32]. *A. halimus* is a halophytic shrub that is widely distributed in arid and semi-arid Mediterranean areas, known for its tolerance to high salinity soils, as well as for exerting an allelopathic effects on other plants [33,34]. It is up to 3 m in height branched from the base, with the bark being grey-white in color and the leaves being 10–30 mm long and 5–20 mm wide. The leaves are highly variable in form, ranging between deltoid-orbicular and lanceolate; these are attenuated at the base with a short petiole. This plant represents a potential source for economical utilization; it can provide forage sources with a good nutritive value during the dry seasons [35,36].

In traditional medicine, *A. halimus* is used to treat a large number of diseases such as inflammation, cracked hands, regulates hormones, heart diseases, diabetes, and rheumatism [37,38]. However, there is yet few works regarding the cytotoxic activities of *A. halimus* specifically against breast cancer cell lines and their antioxidant properties. The crude extract and fractions of *A. halimus* growing in Egypt have cytotoxic activity against MCF-7 and PC3 carcinoma cells and human hepatocellular carcinoma [39]. Moreover, *Atriplex confertifolia* have a cytotoxicity effect on human cervical cancer cells [40].

The aim of the present study was to determine the chemical composition of ethanolic extract from *A. halimus* using LC–MS/MS. The antioxidant and anti-cancer activities were predicted through in silico receptor–ligand docking studies to provide an insight into AHEE’s bioactive compounds’ mechanisms of action. Additionally, the antioxidant activity of *A. halimus* leaf ethanolic extract (AHEE) was performed by five methods, namely, DPPH scavenging capacity, β-carotene bleaching assay, ABTS scavenging, iron chelation, and the total antioxidant capacity assays. Finally, the anti-cancer activity of AHEE was evaluated in human cancer breast cells MCF-7 and MDA-MB-231.

## 2. Results and Discussion

### 2.1. LC–MS/MS Phytochemical Profiling Results

An LC–MS/MS phytochemical investigation was conducted to reveal *A. halimus* ethanolic extract (AHEE) bioactive compounds (Table 1 and Figure 1). The results of this investigation showed an abundance of phenolic acids in the extract, which was mainly presented by the abundance of gallic acid, syringic acid, *trans*-ferulic acid, caffeic acid, and chlorogenic acid. As for flavonoid compounds, we found the presence of myricetin, catechin gallate (a flavan-3-ol), and trimethoxyflavone in lower abundance. Arbutin, which is a glycosylated hydroquinone, was also found in AHEE but in a low amount. In AHEE, gallic acid was the most abundant phytoconstituent, followed by syringic acid and *trans*-ferulic acid.

### 2.2. Pharmacokinetic Properties (ADME) of AHEE

As drug development research has recently known a significant increase in terms of methods, failure of novel molecules as drug candidates remains a substantial issue due to poor pharmacokinetics or bioavailability [47,48]. CADD (computer-aided drug design) represents a time and effort optimizer contributing an alternative in pharmacology, namely, in silico ADME (absorption, distribution, metabolism, and excretion) analysis, which was conducted in this study in furtherance of investigating the bioactive compounds of *A. halimus* leaf ethanolic extract by predicting a broad range of parameters.

Drug likeness of the 10 compounds was evaluated under four filters (Lipinski [49], Ghose [50], Veber [51], Egan [52]) (Table 2). Lipinski and Egan criteria were satisfied for all the molecules except myricetin and catechin gallate, displaying a violation of Lipinski’s rules (NH/OH > 5); hence, a violation displayed for Egan rules regarding myricetin, catechin gallate, and chlorogenic acid was a result of their high polarity (Topological Polar Surface Area (TPSA) exceeding 132 Å^2^). Bioavailability scores were set following an evaluation of six parameters for each molecule counting lipophilicity, molecular weight, insolubility, instauration, and flexibility (Table 2.). *trans*-Ferulic acid was found to have the highest bioavailability score (0.85), due to the fact that the later compound had a TPSA < 75 Å^2^, according to Martin et al. (2005) [53].

In terms of absorption, the molecules were revealed to have a good solubility, which is an essential determinant for drugs’ bioavailability and effectiveness. Caco-2 permeability (given as log Papp in 10^−6^ cm/s) was used for predicting the absorption of a molecule by the intestinal barrier [54]. According to the results above (Table 3), the phytoconstituents did not show a high Caco-2 permeability. Overall, the molecule tended to have good intestinal absorption and a moderate skin permeability. 

As part of absorption prediction, it is mandatory to evaluate whether the molecules are substrate or inhibitors to the most eminent of ABC transporters, P-glycoprotein. Reportedly, no molecule is a substrate to P-glycoprotein, and whereas trimethoxyflavone is an inhibitor of P-glycoprotein I and II, catechin gallate and trimethoxyflavone are inhibitors only for P-glycoprotein II.

Acknowledging drugs’ distribution involves partially considering certain parameters adducing VDss (volume of distribution at steady state), blood–brain barrier permeability, and central nervous system permeability (Table 4 and Figure 2). Overall, the compounds are considered to have a low-to-moderate VDss that can be translated into a good distribution in the plasma. Eventually, the phytocompounds did not show the ability to penetrate the central nervous system.

It is generally relevant to highlight the predicted activity of a molecule and its interactions along with CYP isozymes to anticipate drug metabolism or toxicity [55]. In this study, trimethoxyflavone came out to be a substrate to CYP3A4, while other molecules are non-substrates to CYP2D6 nor CYP3A4. In contrast, inhibition of both CYP2D6 and CYP3A4 was predicted for trimethoxyflavone, whereas myricetin only inhibits CYP3A4 and therefore enhances the risk for adverse effects to occur, including DDI (drug–drug interactions) (Table 5).

Renal clearance liable to excretion is crucial for drug disposition via major organic cation transporters featuring the Renal OCT2, for instance, which has in this case trimethoxyflavone as a substrate [56]. The total clearance investigated for the compounds was measured by reassembling hepatic and renal clearance [57] (Table 6).

### 2.3. In Silico Toxicity Prediction (Using Pro-Tox II)

One of the major drug discovery challenges is the determination of molecule toxic endpoints. In silico predictions represent an efficient alternative that has received considerable interest among drug developers, relying on assessing toxicity profiles on drug candidate molecules [58]. Table 7 displays seven estimated parameters for each molecule, namely, LD₅₀; toxicity class; and their probability to cause hepatotoxic, carcinogenic, immunotoxic, mutagenic, and cytotoxic effects.

As shown in Table 7, the toxicity class was deduced from the median lethal dose, as reported in the GHS (Globally Harmonized System of Classification and Labeling of Chemicals). Myricetin showed the highest value of toxicity among them with 159 mg/kg as a median lethal dose, labelled as class 3 (“toxic if swallowed”). On the other hand, the phytomolecules 1, 2, 3, 5, and 6 were considered “harmful if swallowed” (class 4), representing a LD₅₀ ranging from 300 to 2000 mg/kg, whereas caffeic acid, chlorogenic acid, arbutin, and trimethoxyflavone were “may be harmful if swallowed” in class 5. Additionally, the molecules reflected certain safety by indicating no cytotoxic effect (in HepG2 cell lines) nor a drug-induced liver injury (DILI). However, certain compounds can have limitations in terms of carcinogenicity, immunotoxicity, and mutagenicity; gallic acid, myricetin, and caffeic acid are more likely to be carcinogens; chlorogenic acid and *trans*-ferulic exhibit immunotoxicity; and Myricetin is detected as a mutagenic.

Certain studies suggest that *A. halimus* extracts may exhibit an hepatoprotective effect [59]; thus, other studies have been reporting anticancer properties in many of the compounds cited in AHEE such as caffeic acid, gallic acid, myricetin, and arbutin, among others [60,61,62,63,64]. Flavonoids have been mentioned to be involved in diverse beneficial activities for organisms but tend to reveal some risk, such as different or adverse properties under certain sets of conditions such as mutagenicity or carcinogenicity [65,66].

### 2.4. In Silico Prediction of a Protein-Target-Based Antioxidant and Cytotoxic Mechanisms by Molecular Docking Analysis

#### 2.4.1. In Silico Prediction of the Antioxidant Activity of AHEE

As it can predict the conformation of small-molecule ligands within the right target binding site with a high degree of accuracy, molecular docking is one of the most widely employed techniques in structure-based drug design (SBDD) [67]. Molecular docking (MD) became a crucial method in drug discovery since the creation of the initial algorithms in the 1980s [68]. For instance, it is simple to carry out experiments involving critical molecular events, such as ligand-binding modalities and the associated intermolecular interactions that stabilize the ligand–receptor complex [69]. MD was used in this study to uncover the possible mechanism of action of AHEE components. Assuming that binding energy decreases as compound affinity increases, the obtained results, in the form of binding affinity values, may indicate an increased or decreased affinity of the researched molecule toward the specified target in comparison to a native ligand (a known inhibitor).

Their molecular docking interactions have been explored with particular enzymatic proteins, such as lipoxygenase-3 (PDB ID: 1N8Q) [70], cytochrome P450 (PDB ID: 1OG5) [71], NADPH oxidase (PDB ID: 2CDU) [72], and bovine serum albumin (PDB ID: 4JK4) [73], all of which have been identified as target receptors for antioxidant chemicals and are acknowledged.

The docking scores were presented in a heat-map-style table with a red, white, and green color scheme, ranging from the lowest energy values, highlighted in red (most often corresponding to the native ligand’s docking score), to the highest, highlighted in green (Figure 3). This made it simple to identify a group of chemicals that frequently act as potential inhibitors by comparing their lowest values to the native ligand for a particular target. Ligands with a docking score lower or equal to the native ligand’s score were highlighted with a star (*).

Using a redox mechanism, the family of enzymes known as lipoxygenases catalyzes the lipid peroxidation of poly-unsaturated free fatty acids. This process results in the production of an oxygen-centered fatty acid hydroperoxide radical, which can cause a number of harmful illnesses [74]. With respect to this, we selected the two lipoxygenases, lipoxygenase (1N8Q) and cytochrome P450 (1OG5). For the first target, all the ligands were found to be potent inhibitors, except syringic acid with −5.7 kcal/mol, compared with the native ligand (protocatechuic acid) with −6.0 kcal/mol (Table 2). The potent inhibitor found in AHEE was catechin gallate with −8.8 kcal/mol, found to establish five hydrogen bonds (HB) with the active site amino acid residues (ASN A:375, ARG A:378, ASP A:597, and two HB with GLN A:598) (Appendix A). For the second target, in comparison to warfarin, which had −6.6 kcal/mol (a natural ligand for CYP2C9), two compounds were identified to be potent inhibitors: myricetin and catechin gallate (with −8.4 kcal/mol for both molecules). The binding research showed that both substances created a hydrogen bond with the amino acids GLN A:214 and ASN A:217 from the active site pocket [75] (Appendix A).

It is possible that AHEE compounds have little ability to inhibit NADPH oxidase protein since none of the docked molecules exhibited free binding energies greater or equal to 8.6 kcal/mol (of the native ligand adenosine-5′-diphosphate).

All the identified compounds in AHEE were found to operate as natural bovine serum albumin (BSA, PDB ID: 4JK4) protein inhibitors, as shown by their interaction scores. Furthermore, myricetin, catechin gallate, and chlorogenic acid were the three most effective ligands, in comparison with the natural ligand of BSA, 3.5-diiodosalicylic acid (−5.3 kcal/mol), at −8.5, −8.4, and −8.3 kcal/mol, respectively. Binding interactions with both NADPH oxidase protein and BSA are found in Appendix A.

The natural ligands’ ability to bind to amino acid residues may be related to their antioxidant activity. Thus, it can be said that the examined compounds have strong antioxidant activity according to an examination of common residues and free binding affinity values near the docked ligands and natural inhibitors, as well as being stated in the literature [75,76].

#### 2.4.2. In Silico Prediction of the Cytotoxic Potential of AHEE

Each of the nine identified compounds’ possible targets were predicted using the SwissTarget Prediction tool [77]. The SMILES strings of the compounds were inputted to the database, and prediction results were reported in Section 3.4.2. Molecular docking was utilized after choosing three targets (that had the best likelihood of being a particular target according to interaction occurrence probability) for each molecule.

Carbonic anhydrases enzymes (CA, EC 4.2.1.1) catalyze reversible carbon dioxide hydration to bicarbonate and protons, which is a very straightforward yet crucial physiological event. In mammals, at least 16 different α-CA isoforms have been discovered. These isozymes have highly distinct catalytic characteristics as well as a very different distribution in various tissues and organs [78]. Recent studies show that these CAs have been linked to chemoresistance, tumor cell motility and invasion, and the maintenance of cancer cell stemness [79,80]. Our prediction results showed that the majority of the identified components are possible inhibitors for three cytosolic isoforms CA I, II, and VII. Our docking analysis for the first isoform of CA I (PDB ID: 1AZM) revealed three possible inhibitors with a docking score higher than the native ligand (furosemide, −6.8 kcal/mol), namely, myrcetin, catechin gallate, and chlorogenic acid with −8.6, −8.4, and −7.4 kcal/mol, respectively (Figure 4). For the second isozyme CA II (PDB ID: 12CA), all the tested molecules were found to be strong inhibitors (in comparison with the native ligand, herein, tetrazole, which has a docking score of −4.2 kcal/mol), with the highest affinity found with chlorogenic acid (−7.2 kcal/mol), followed by caffeic acid, gallic acid, *trans*-ferulic acid, and syringic acid with binding free energy values of −6.4, −5.6, −5.5, and −5.4 kcal/mol, respectively. The third isozyme, which was CA VII (PDB ID: 3MDZ), found abundantly in the central nervous system, characterized by its high CO_2_ hydration activity and linked to various illnesses such as epilepsy, oxidative stress, and cancer [81,82], was used as a target for six molecules from AHEE. The results showed that three molecules out of six (chlorogenic acid, arbutin, and caffeic acid, with −8.4, −6.5, and −6.4, respectively) had an affinity greater than that of the native ligand (acetazolamide, −6.3 kcal/mol) (Figure 4).

Human milk xanthine oxidoreductase (XOR) is a form of xanthine oxidoreductase, a homodimer of ≈300 kDa, that ensures the catabolism of hypoxanthine and xanthine to uric acid [83]. In low-XOR-expressing cancer cells, the amount of XOR expression may be linked to a worse prognosis due to the inflammatory response brought on by the tissue damage caused by tumor growth. Because it can catalyze the metabolic activation of chemicals that cause cancer, xanthine oxidoreductase (XOR) has been linked to oncogenesis either directly or indirectly by reactive oxygen and nitrogen species that are produced by XOR [84]. The 3D crystal structure of XOR (PDB ID: 2CKJ) was used a as target of myricetin, and the docking score of this molecule (−8.6 kcal/mol) was found to be lower than that of oxypurinol (−6.2 kcal/mol) (Figure 4). These results are demonstrated by a previous study by Zhang et al. (2016) [85], providing experimental data on the role of myricetin in inhibiting the activity of XOR.

Myricetin was found to be a potent inhibitor (with −9.6 kcal/mol in comparison with the another inhibitor at −5.2 kcal/mol) for human DNA topoisomerases IIa, crucial DNA topoisomerases that have crucial roles in chromosome segregation and chromatine condensation, having been proven to be therapeutic targets of anticancer drugs [86,87].

Catechin gallate was found to be a specific ligand for two cancer-implicated targets, human phosphogluconate dehydrogenase (PGD, PDB ID: 2JKV) and human antiapoptotic protein BCL-2 (PDB ID: 1G5M), with docking scores of −9.2 and −7.5 kcal/mol, respectively, in comparison with their respective native ligands (−6.6 and −5.3 kcal/mol) (Figure 4). The first target PGD catalyzes the conversion of 6-phosphogluconate (6-PG) into ribulose 5-phosphate (R-5-P) in the third phase of the pentose phosphate pathway (PPP), resulting in the production of nicotinamide adenine dinucleotide phosphate (NADPH) [88]. There have been numerous reports of 6PGD upregulation in human malignancies [89]. Therefore, focusing on PGD may be an alluring way to fight a fatal illness such as cancer. For the second target (BCL-2), it has been shown that many different kinds of cancer exhibit considerably increased BCL-2 protein expression [90]. Instead of encouraging proliferation, BCL-2 promotes the survival of cancer cells. This concept has helped in realizing that a damaged or disturbed apoptotic pathway might be extremely important in the genesis of tumors. Since BCL-2’s oncogenic potential has been established and is well known, it is thought to be a viable target in the therapy of cancer [91].

Adenosine deaminase (ADA, E.C 3.5.4.4) catalyzes the irreversible deamination of both adenosine and 2′deoxyadenosine. Given that ADA activity controls the pool of intracellular and extracellular adenosine, a crucial regulator of cellular function via adenosine-receptor-dependent and -independent processes, its role in the development of breast cancer appears to be particularly significant. It has been demonstrated that the levels of ADA1 and ADA2 isoenzymes in breast cancer tumor tissues were raised and correlated with the tumor’s grade, size, and lymph node involvement [92,93]. Since ADA plays a major role in DNA turnover and nucleotide metabolism, ADA inhibitors have been extensively used for chemotherapeutic purposes in some types of cancers including breast cancer [94,95]. For this regard, arbutin was discovered to be a powerful inhibitor with a docking score of −7.4 kcal/mol compared to the native ligand (2′-deoxyadenosine) with −6.2 kcal/mol (Figure 4). Purine nucleoside phosphorylase (PNP, PDB ID: 1M73) was also identified as a target of arbutin, but its activity as a potential inhibitor was insignificant, which suggests that the cytotoxic activity of AHEE is not mediated via PNP inhibition and immune activation [96].

A_1_, A_2A_, A_2B_, and A_3_ are four G-protein-coupled receptor subtypes that interact with adenosine to mediate its physiological effects. Adenosine appears to be associated with the development of tumors; nevertheless, it has been noted that cancer tissues have significant amounts of the substance [97]. A_1_ receptor (PDB ID: 5N2S) was found as a target for trimethoxyflavone. The latter was found to be a strong inhibitor. Docking score of trimethoxyflavone was −7.9 kcal/mol, while that of PSB36 (1-butyl-3-(3-hydroxypropyl)-8-(3-noradamantyl) xanthine, a potent antagonist of A1 receptor, was −6.0 kcal/mol (Figure 4). The A_2A_ and A_3_ subtypes of adenosine receptors appear to be the most promising in terms of drug development, despite the fact that all adenosine receptors now have an increasing number of biological roles in cancers that are recognized. A_2A_ receptor activation in particular has immunosuppressive effects that lower anti-tumor immunity and promote tumor development. This tendency has led to the suggestion that A_2A_ antagonists be added to cancer immunotherapeutic procedures in order to improve tumor immunotherapy. In this regard, trimethoxyflavone was regarded as an inhibitor due to its high affinity (−9.0 kcal/mol) with A_2_ receptor (PDB ID: 2YDO), in comparison with a known ligand (istradefylline, −8.4 kcal/mol) (Figure 4).

The inherent or acquired resistance of tumor cells to chemotherapeutic drugs with different chemical structures and modes of action is one of the major reasons why treatment for many malignancies fails [97]. One of these processes entails the activation or overexpression of drug-export proteins called ATP-binding cassette (ABC) protein transporters, which lowers the levels of drug accumulation in the cell [98,99]. ABC transporter (PDB ID: 5NJ3) was found to be a specific target of trimethoxyflavone, but the latter substance had lower affinity (−8.0 kcal/mol) when compared to gefitinib (−8.9 kcal/mol) (Figure 4), suggesting that AHEE’s mechanism of action is not mediated by ABC transporter inhibition. 

At different stages of cancer development, phenols were found to exhibit various anticancer mechanisms. These processes include inhibition of DNA-related enzymes including topoisomerase, blockage of the estrogen receptor, obstruction of cell cycle, and cell death [100,101]. These findings suggest that AHEE’s bioactive compounds may exhibit a potent anticancer activity.

### 2.5. Experimental Validation of the Antioxidant and the Cytotoxic Activity of AHEE

#### 2.5.1. Antioxidant Activity

The determination of the antioxidant properties of *A. halimus* ethanolic extract (AHEE) was performed using five methods: DPPH, β-carotene, ABTS, iron chelation, and total antioxidant assays, and the results are summarized in Table 8. The results obtained showed significant antioxidant activities of AHEE. 

The AHEE exhibited the highest capacity to reduce the purple-colored solution of DPPH radical to yellow-colored non-radical form DPPH-H with an IC_50_ value equal to 0.36 ± 0.05 mg/mL when compared with the antioxidant ascorbic acid (0.19 ± 0.02 mg/mL). In this sense, it was reported that butanolic and ethyl acetate fractions of *A. halimus* showed stronger DPPH scavenging activity with EC_50_ values of 1.73 (R_2_ = 0.98) and 2.04 (R_2_ = 0.99) mg/mL, respectively [36]. 

A promising β-carotene-bleaching activity was observed also with the AHEE (2.91 ± 0.14 mg/mL) compared to the reference BHA (0.095 ± 0.00 mg/mL). Furthermore, the results obtained by the ABTS method were found to scavenge ABTS with an IC_50_ value of 44.10 ± 2.92 TE µmol/mL. However, this value was less than that of the standard BHA (IC_50_= 5.04 ± 0.78 mg/mL). On the other hand, metals ions are important for the functioning of physiological cellular processes but, at the same time, the iron overload induced the formation of lipid peroxidation products, which have been demonstrated in a number of tissues, such as the brain, liver, and kidneys [102,103]. Nevertheless, the ethanolic extract showed a remarkable antioxidant in the iron chelation assay, with an IC_50_ value equal to 27.40 ± 1.46 mg/mL. This value was also higher than that of the ascorbic acid (0.94 ± 0.02 mg/mL). Moreover, the iron chelators, on interacting with DPPH, might have transferred an electron to it, thus neutralizing its free radical nature, as observed by Oyaizu [104]. 

The total antioxidant capacity (TAC) IC_50_ value for AHEE was 124 ± 1.27 µg AA/mg extract. The value of the total antioxidant capacity of AHEE found in this study may point to the fact that AHEE is very rich in natural antioxidant compounds. Our results are consistent with the results obtained by Bouaziz et al. (2021) [105]. In the same way, Benhammou et al., (2009) showed that the methanolic extract of *A. halimus* exhibited a potent antioxidant activity [36]. 

A study conducted by Slama et al., (2020) showed that this plant possesses an important scavenging activity [35]. Previous findings report this antioxidant capacity to the presence of polyphenols, flavonoids, and tannins [35,36]. It is well known that flavonoids are known to have numerous several biological properties, including anticancer and anticarcinogenic activities. These pharmacological effects that they exhibit could be attributed to their ability to effectively scavenge or chelate reactive oxygen species [106]. Dorman reported that plant flavonoids that induce an antioxidant capacity in vitro correspondingly function as antioxidants in vivo [107]. 

Previous reports showed several phenolic acids such as gallic acid, chlorogenic acid, cinnamic acid, *p*-coumaric acid, hydroxybenzoic acid, ferulic acid, and salicylic acid [108]. Likewise, previous findings reported different flavonoid compounds present in *A. halimus* extract such as quercetin, rutin, and myricetin [108,109]. These results have indicated that *A. halimus* is a potential candidate of antioxidant activities.

#### 2.5.2. Cytotoxicity of AHEE against Breast Cancer Cell Lines (MCF-7 and MDA-MB-231)

Numerous ethnobotanical studies reported the use of *A. halimus* in the treatment of cancer [110,111,112]. In addition, in vitro studies showed that water extracts of *A. halimus* had a very interesting activity against different cancer cell lines [112,113]. In the present study, we investigated the antiproliferative activity of *A. halimus* ethanolic extract toward MCF-7 and MDA-MB-231 human breast cancer cell lines, using the MTT assay. Figure 5 shows the effect of increasing concentrations of AHEE and cisplatin on the tumor cells. As can be seen, AHEE showed a dose-dependent effect toward both tumor cell lines, where MCF-7 cancer cell line had the highest sensitivity. Table 1 shows the IC_50_ values and the selectivity indexes. It is depicted in this table that AHEE exhibited a potent antitumor activity on MCF-7 and MDA-MB-231 with IC_50_ values equal to 27.85 ± 3.14 µg/mL and 51.95 ± 7.03 µg/mL, respectively. Cisplatin was used as a positive control and rendered IC_50_ values of 3.66 ± 1.05 µg/mL and 1.60 ± 1.19 µg/mL, respectively. Our results were expected to be better compared to various research works such as those of Hosny et al. (2021) [113] and Al-Senoy et al. (2018) [39], who examined the efficacy of *A. halimus* extract on MCF-7 and reported moderate efficacy with IC50 of 47.03 µg/mL and 153.6 µg/mL, respectively.

A minimum level of cytotoxicity is required from anticancer drug candidates since blood cells are the first normal cells to be exposed to chemotherapeutic agents. Therefore, we evaluated the effect of AHEE on PBMC viability using the MTT assay. Interestingly, AHEE showed very low suppressive activity against human PBMCs, with an IC_50_ value above 600 µg/mL. AHEE showed the highest selectivity for MCF-7 cells with a safety ratio of 26.7, followed by MDA-MB-231 cells with a safety ratio of 14 (Table 9). However, cisplatin showed a very low safety ratio towards MCF-7 cells (8.15) compared to MDA-MB-231 cells (18.64). In effect, these results revealed that AHEE is more selective to cancer cell lines and might not affect normal cells. Indeed, these results are in agreement with the work of Al-Senoy et al. (2018) [39], who revealed that AHEE is more selective towards cancer cell lines and might not affect normal cells.

## 3. Materials and Methods

### 3.1. Plant Material and Extraction

*Atriplex halimus* (L.) was collected and then identified in the Department of Biology, University Mohammed First Oujda, Morocco. A voucher specimen was deposited at the herbarium of the same faculty under the number HUMPOM543. Leaves were shade dried, grounded, powdered (using a grinding mill, model SM-450, MRC., Ltd., Changzhou, China), and extracted with ethanol (70%). Extracts were dried under vacuum in a rotary evaporator (Model RE501, Yuan Huai Chemical Technology Co., Ltd., Shanghai, China) to produce the crude extract.

### 3.2. LC–MS/MS Profiling of AHEE

Aliquots of the samples (80 mg) were extracted using the procedure described in [114,115]; below is the procedure in detail: 1 mL of ethanol was added to the aliquot. The Eppendorf tube was vortexed and submerged for 60 min in a 45 °C sonicator bath. A Shimadzu Ultra-High-Performance Liquid Chromatograph (UHPLC, Nexera XR LC 40, Kyoto, Japan) with an MS/MS detector was used for the qualitative analysis (LCMS 8060, Shimadzu Italy, Milan, Italy). The Lab Solution software (ver. 5.6, Kyoto, Japan) was used to manage the MS/MS, which used electrospray ionization. This program allowed for a quick transition from a low energy scan of 4 V (full scan MS) to a high energy scan (10–60 V ramping) during a single LC cycle. A total of 2.9 L/min nebulizing gas flow, 10 L/min heating gas flow, 300 °C interface temperature, 250 °C DL temperature, 400 °C heat-block temperature, and 10 L/min drying gas flow were the settings chosen for the source. The analysis used a mobile phase of acetonitrile and water + 0.01 percent formic acid in a 5:95 (*v*/*v*) ratio, with no chromatographic separation. Mass spectral data acquisition was achieved under negative electrospray ionization (ESI-). A molecule was considered positive if its area under the curve was greater than that of the blank (information on the molecules’ retention times and typical fragments in m/z are available in Appendix A; typical fragments of the used standards are found in Appendix A). The molecules were identified by comparing the typical fragment identified with those in our in-house-developed library of molecules.

### 3.3. Prediction of the Pharmacokinetic Properties and Toxicity of AHEE’s Bioactive Compounds

ADME (absorption, distribution, metabolism) profiles were assessed in silico via computational tools using the SwissADME online server (http://www.swissadme.ch/, accessed on 25 July 2022) [116] and pkCSM web-server (http://biosig.unimelb.edu.au/pkcsm/, accessed on 25 July 2022) [57,117]. The estimation of LD₅₀ values, toxicity class, hepatotoxicity, carcinogenecity, immunotoxicity, mutagenicity, and cytotoxicity was carried out by the Protox II online tool (https://tox-new.charite.de/protox_II/, accessed on 28 July 2022) [118].

### 3.4. Molecular Docking Prediction of the Antioxidant and Cytotoxic Activity of AHEE

#### 3.4.1. Molecular Docking General Procedure

Using the target proteins’ PDB IDs, the crystallographic three-dimensional structures of the target proteins were obtained from the protein data bank (https://www.rcsb.org/) (accessed on July 25 July 2022) and displayed using the Discovery Studio 4.1 (Dassault Systems Biovia, San Diego, CA, USA) program. With the same tool, typical inhibitors, water molecules, and ions were also eliminated. Automated docking investigations were conducted using AutoDock Vina v1.5.6 software [119]. The protein also acquired polar hydrogen bonds and Kollmann charges [120]. The identified compounds in AHEE were retrieved from https://pubchem.ncbi.nlm.nih.gov/ (accessed on 25 July 2022) as “3D sdf” format and then converted using PyMol to pdb file format.

Using MGL tools, three-dimensional PDBQT files of the protein and ligands were created. In order to generate the grid maps with the interaction energies on the basis of the macromolecule target of the docking investigation, AutoGrid grid, a part of Auto Dock, was employed.

The grid box used to represent the docking search space was expanded to accommodate the current binding site more accurately. Kcal/mol values for the ΔG binding energies were used to express the findings for the docked ligand complexes. Using Discovery Studio 4.1, 2D molecular interaction schemes were created and protein–ligand binding interactions were examined. On the basis of the literature, we selected four antioxidant proteins for the prediction of antioxidant activity: lipoxygenase (PDB: 1N8Q), CYP2C9 (PDB: 1OG5), NADPH oxidase (PDB: 2CDU), and bovine serum albumin (PDB: 4JK4).

#### 3.4.2. Ligand-Based Target Prediction with SwissTarget Prediction

In the area of drug development, ligand-based target prediction has been proven to be extremely effective and quick at predicting the proper protein targets of compounds [121]. The “molecular similarity hypothesis”, which postulates that similar chemicals target common proteins, was validated by measuring the similarity of compounds using a variety of techniques [122]. In this study, SwissTarget Prediction (https://www.swisstargetprediction.ch) (accessed on the 24 July 2022), a web application that performs ligand-based target prediction for any bioactive small molecule and has been available since 2014 [77], was used to predict breast-cancer-related protein targets, and three of the targets found for each compound with the highest probability were chosen. Table 10 summarizes the results of our analysis.

### 3.5. Anticancer Activity

#### 3.5.1. Cell Culture

MCF-7 ER-positive and MDA-MB-231 ER-negative breast cancer cells were used. Dulbecco’s minimum essential medium (DMEM) with 10% fetal bovine serum (FBS) and 50 µg/mL gentamicin was used to cultivate the cells. The cells were incubated in a humidified atmosphere at 37 °C and 5% CO_2_. The cells were kept alive by subculturing them in 25 cm^2^ tissue culture flasks. For the cell viability experiment, cells in the exponential phase were employed.

#### 3.5.2. Cell Viability by MTT Assay

The MTT assay was performed to determine whether AHEE inhibited cancer cell proliferation according to the method described in [134]. MCF-7 and MDA-MB231 cells that were exponentially proliferating were plated onto 96-well plates (10^4^ cells per well in 100 µL of medium) and left to attach for 24 h. To attain acceptable concentrations, the AHEE were solubilized in 0.1% DMSO and serially diluted with medium. Cells were treated with AHEE at various doses and incubated for 72 h. Cells in the control group received only 0.1% DMSO-containing medium. The test compound media was replaced with 200 µL of culture medium before adding 20 µL of MTT reagent (5 mg/mL MTT in PBS) and incubating for 4 h at 37 °C. The medium was withdrawn, and 100 µL of DMSO was added before absorbance was measured at 540 nm with a microplate reader (Synergy HT Multi-Detection microplate reader, Bio-Tek, Winooski, VT, USA) and percentage viability was calculated [135].
$$\mathrm{Cell}\;\mathrm{viability}\;\left(\%\right)=100-\left[\left(\frac{A_{0}-A_{t}}{A_{0}}\right)\times 100\right]$$
where A_0_ = absorbance of cells treated with 0.1% DMSO medium, A_t_ = absorbance of cells treated with AHEE at different concentrations. A total of 0.1% (*v*/*v*) DMSO in medium was used as the negative control. Each treatment was performed in triplicate. Cisplatin was used as standard. IC_50_ values were calculated using dose–response inhibition curves in Graph pad prism 8.01.

Following approval by the Research Ethics Committee (03/21-LAPABE-10 and 4 March 2021), PBMCs were isolated from human blood samples by Ficollhypaque density centrifugation according to the manufacturer’s instructions (Capricorn Scientific). The cytotoxic effect was evaluated under the same conditions and concentrations as previously described for tumor cells.

### 3.6. Antioxidant Activity 

#### 3.6.1. 2,2-Diphenyl-1-Picrylhydrazil Free Radical Scavenging Assay

The determination of the free radical scavenging capacity of AHEE was determined according to the methods described in [75,136], with modifications. DPPH solution was prepared by solubilizing 2 mg of DPPH in 100 mL of methanol. Different concentrations ranging from 5 to 500 µg/mL were prepared. Afterwards, each concentration was added to 2.5 mL of the prepared DPPH methanol solution to the final volume of 3 mL. After 30 min of incubation at room temperature, the absorbance was measured at 515 nm against a blank. DPPH free radical scavenging activity as a percentage (%) was calculated using the following formula:$$\mathrm{Radical}\;\mathrm{Scavenging}\;\mathrm{Activity}\;\left(\%\right)=\left[\left(\frac{{A\;}_{\mathrm{blank}}-{A\;}_{\mathrm{sample}}}{{A\;}_{\mathrm{blank}}}\right)\right]\times 100$$
where A blank is the absorbance of the control reaction (all reagents except the extract are present) and A sample is the absorbance of the extract at different concentrations. The IC_50_ was obtained by plotting the inhibition percentage versus extract concentrations on a graph. Ascorbic acid was employed as a positive control.

#### 3.6.2. β-Carotene Bleaching Assay

The antioxidant activity was performed using bleaching of a β-carotene assay [75,137]. Briefly, 2 mg of β-carotene was solubilized in 10 mL of chloroform before being added to 20 mg of linoleic acid and 200 mg of Tween-80. Following the removal of the chloroform mixture by rotavapor at 40 °C, 100 mL of distilled water was added to the flask with vigorous agitation. Afterward, all samples were placed in triplicate in a 96-well plate and kept in the dark at 25 °C for 30 min. Thereafter, absorbance was measured spectrophotometrically at 470 nm immediately after AHEE solution addition (t0) and after two hours of incubation (t1), both against a white reading containing all of the previous AHEE solution components but no-carotene. BHA was used as a standard reference.
$$\mathrm{Residual}\;\mathrm{color}\;\left(\%\right)=\left[\left(\frac{\mathrm{Initial}\;\mathrm{OD}\;-\mathrm{Sample}\;\mathrm{OD}}{\mathrm{Initial}\;\mathrm{OD}}\right)\right]\times 100$$

#### 3.6.3. ABTS Scavenging Activity Assay

The scavenging capacity to the 2,2-azino-bis-(3-ethylbenzothiazoline-6-sulfonic acid) (ABTS) radical of AHEE was investigated, as described by Nakyai et al. [138], with some modifications. To create ABTS•+, the ABTS solution was combined with 2.45 mM potassium persulfate and incubated at room temperature for 16–18 h in the dark. The solution was then diluted with ethanol to achieve an absorbance of 0.70 ± 0.02 at 750 nm. In ethanol, an extract stock solution was produced. As a positive control, L-ascorbic acid was employed. The ABTS assay was carried out by combining 200 µL of diluted ABTS•+ solution with 20 µL of test sample. This reaction mixture was incubated in the dark at room temperature for 10 min before being measured with a microplate reader at 734 nm. In a manner comparable to the DPPH assay, a percentage of ABTS radical cation decolorizing activity was calculated.

#### 3.6.4. Iron Chelation

According to the protocol described by Chaudhary et al. (2015) [134], in 100 mL of distilled water, 0.198 g of 1,l0-phenanthroline monohydrate, 2 mL of 1 M hydrochloric acid, and 0.16 g of ferric ammonium sulfate were mixed to make the 1,l0-phenanthroline-iron (III) reagent. Concisely, 0.2 mL of standard/extracts were combined with 0.2 mL of 1,l0-phenanthroline-iron (III) reagent, 0.6 mL of methanol, and 4 mL of water. After 30 min of incubation at 50 °C, the absorbance was measured at 510 nm. Positive control was ascorbic acid. Higher absorbance meant more iron chelating activity. The following formula was used to compute percentage scavenging:$$\mathrm{Scavenging}\;\mathrm{Activity}\;\left(\%\right)=\left[\left(\frac{A\;\mathrm{sample}-A\;\mathrm{control}}{A\;\mathrm{control}}\right)\right]\times 100$$
where A control = absorbance of control (without extract) and A sample = absorbance of sample.

#### 3.6.5. Total Antioxidant Capacity

The total antioxidant activity was determined using the phosphor-molybdenum method as described in [134]. Incubated at 95 °C for 90 min, 0.1 mL of standard/extract solution was combined with 0.3 mL of reagent solution (0.6 M sulfuric acid, 28 mM sodium phosphate, and 4 mM ammonium molybdate). The mixture was cooled to room temperature, and the absorbance at 695 nm was measured. Except for the test sample, the blank solution contained all of the reagents. The standard curve was created using ascorbic acid. The findings were reported in terms of ascorbic acid equivalents [139]. All experiments were carried out in triplicate.

## 4. Conclusions

The results obtained in the present study showed a richness of the phytochemical profile of AHEE in terms of phenolic compounds such as gallic acid, syringic acid, and *trans*-ferulic acid, among others. The results of the antioxidant activity have indicated that AHEE may be considered as a potent antioxidant candidate. For the cytotoxicity assays, two human breast cancer cell lines (MCF-7 and MDA-MB-231) were studied throughout this work to better understand the cytotoxicity impact of AHEE on cancer cells. AHEE was found to have substantial cytotoxic effects against breast cancer cell lines, particularly the estrogen receptor (ER)-positive cell line MCF-7. The cytotoxic activity analysis revealed substantial cytotoxic effects in a dose-dependent manner against both breast carcinoma cell lines (MCF-7 and MDA-MB-231). However, AHEE did not appear to be cytotoxic to normal cells (PBMCs). According to the findings, AHEE may provide a healthy and natural source of bioactive substances that may be utilized both preventively and in the clinic without causing toxicity. 

## Figures and Tables

**Figure 1 pharmaceuticals-15-01156-f001:**
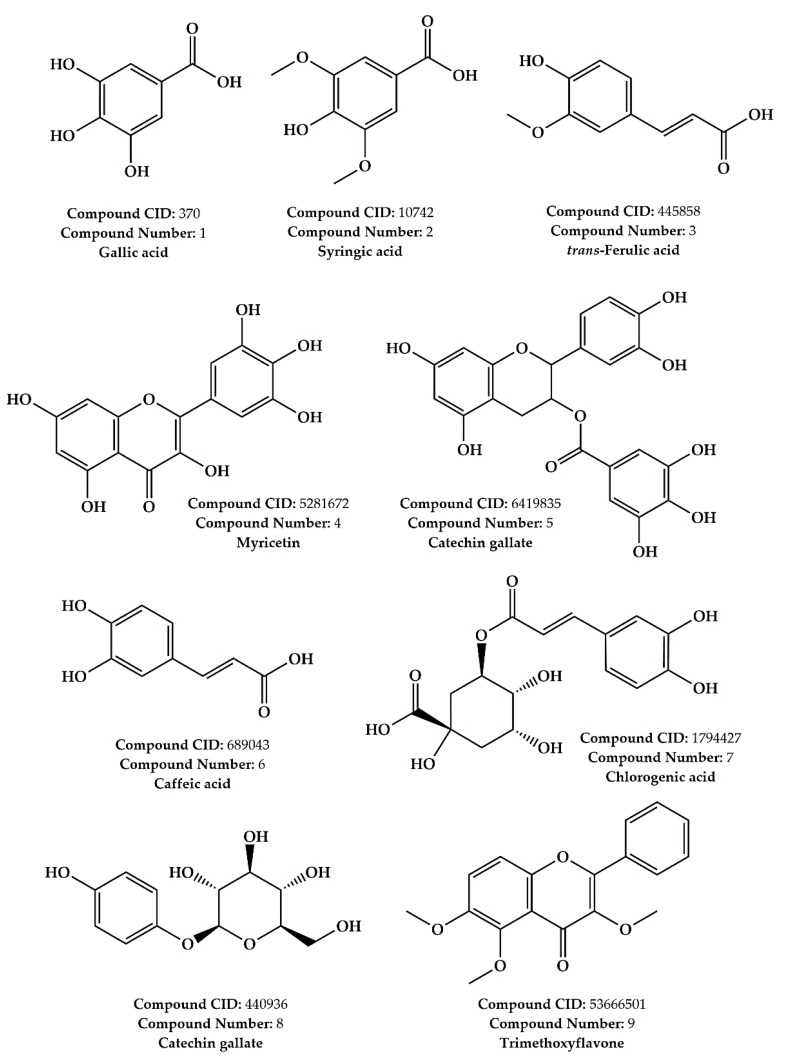
Chemical structures of the identified compounds in *A. halimus* ethanolic extract using LC–MS/MS.

**Figure 2 pharmaceuticals-15-01156-f002:**
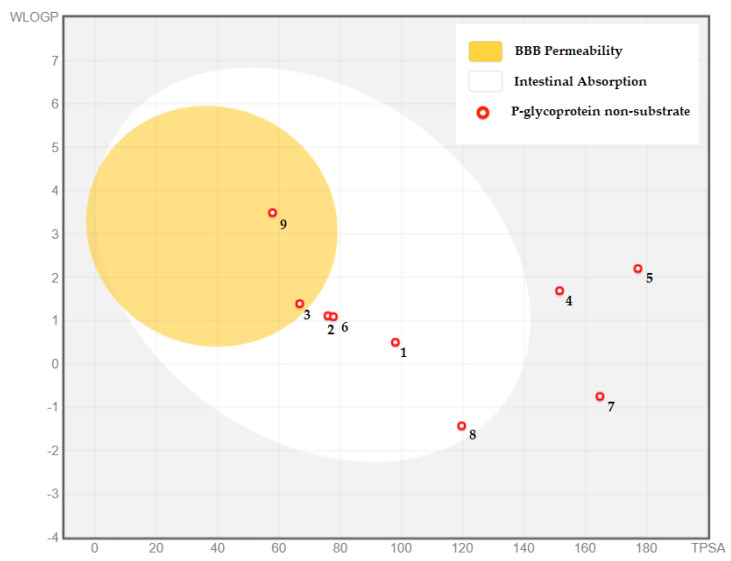
BOILED-Egg method for evaluating blood–brain barrier permeability, gastrointestinal absorption, substrates, and inhibitors of P-glycoprotein for *A. halimus* leaf chemical composition. (**1**) Gallic acid, (**2**) syringic acid, (**3**) *trans*-ferulic acid, (**4**) myricetin, (**5**) catechin gallate, (**6**) caffeic acid, (**7**) chlorogenic acid, (**8**) arbutin, (**9**) trimethoxyflavone.

**Figure 3 pharmaceuticals-15-01156-f003:**
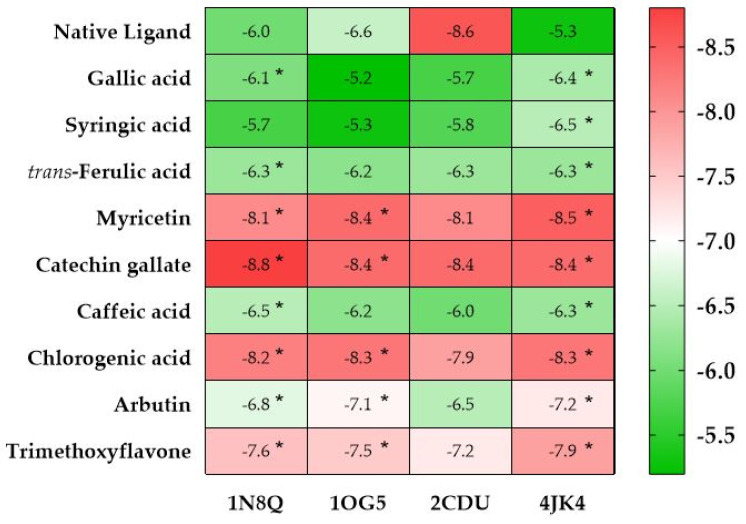
Heat map of the binding free energy values (expressed in kcal/mol) of *A. halimus* ethanolic extract phytoconstituents. **1N8Q**, lipoxygenase; **1OG5**, CYP2C9; **2CDU**, NADPH oxidase; **4JK4**, bovine serum albumin. Ligands with a docking score lower or equal to the native ligand’s score were highlighted with a star (*).

**Figure 4 pharmaceuticals-15-01156-f004:**
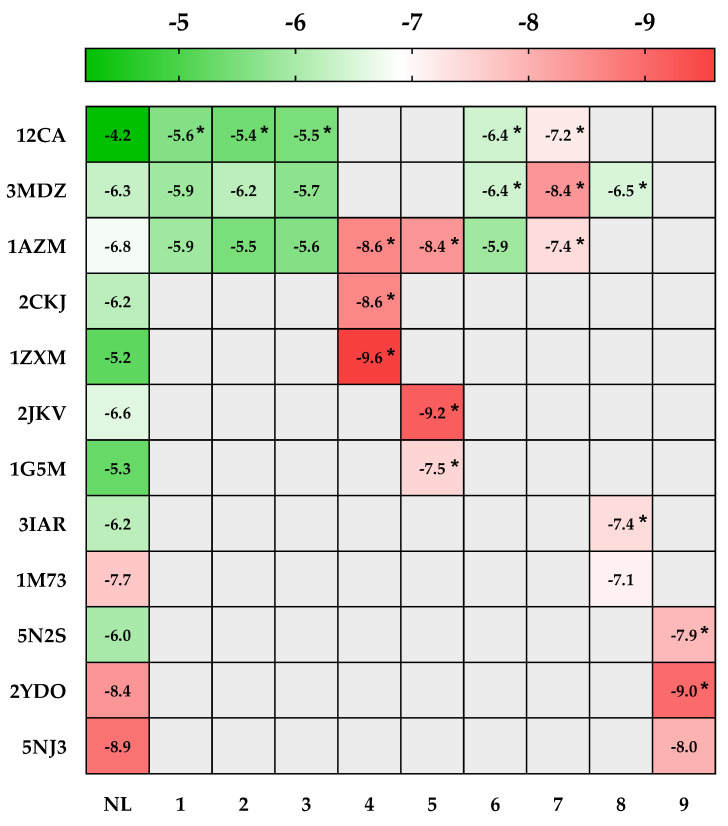
Heat map of the binding free energy values (expressed in kcal/mol) of *A. halimus* ethanolic extract phytoconstituents. Left axis shows the predicted targets (see Section 3.4.2). (**1**) Gallic acid, (**2**) syringic acid, (**3**) *trans*-ferulic acid, (**4**) myricetin, (**5**) catechin gallate, (**6**) caffeic acid, (**7**) chlorogenic acid, (**8**) arbutin, (**9**) trimethoxyflavone. NL refers to the native ligands (Section 3.4.2). Ligands with a docking score lower or equal to the native ligand’s score were highlighted with a star (*).

**Figure 5 pharmaceuticals-15-01156-f005:**
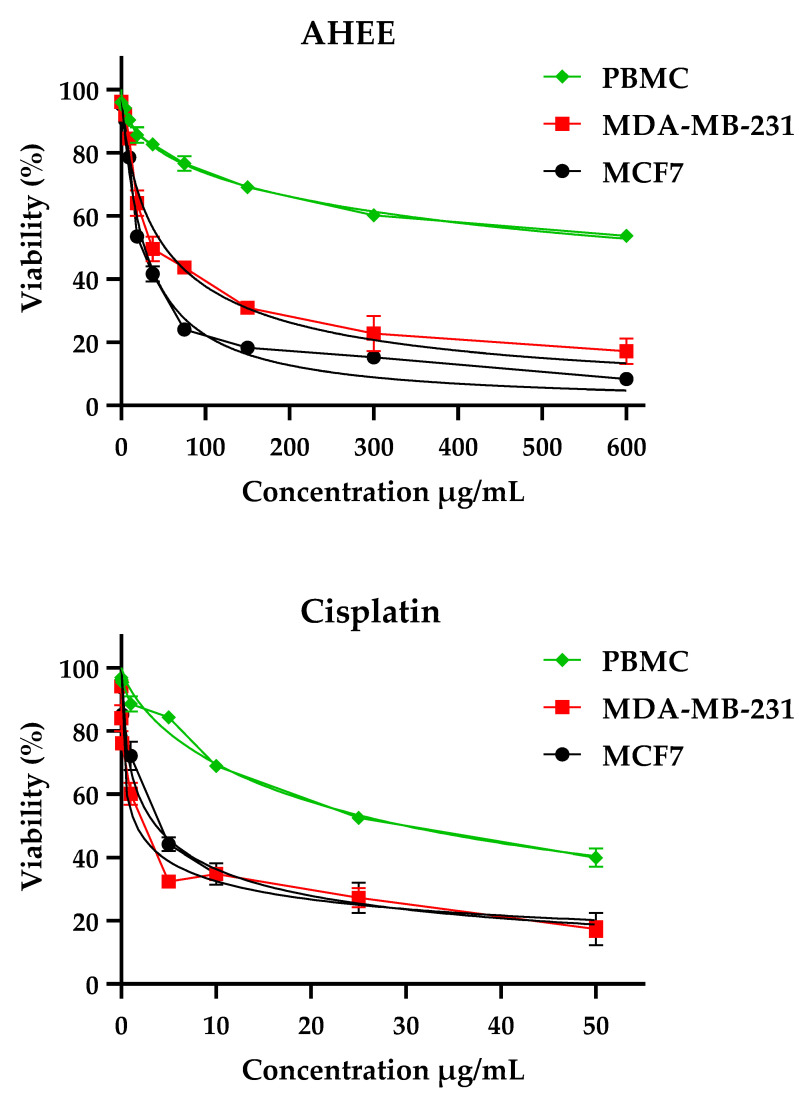
Cell viability of MCF-7, MDA-MB-231, and PBMC cells after 72 h of treatment with *A. halimus* ethanolic extract and cisplatin using MTT test.

**Table 1 pharmaceuticals-15-01156-t001:** Phenolic composition of *A. halimus* ethanolic extract with LC–MS/MS.

N°	Molecule	Molecular Formula	Selected [M−H]^−^	Literature [M−H]^−^	RT (min)	Abundance
1	Gallic acid	C_7_H_6_O_5_	168.90	169.00 [41]	1.586	+++
2	Syringic acid	C_9_H_10_O_5_	198.90	197.05 [42]	1.478	++
3	*trans*-Ferulic acid	C_10_H_10_O_4_	193.00	193.05 [42]	1.191	++
4	Myricetin	C_15_H_10_O_8_	317.00	317.01 [43]	0.330	+
5	Catechin gallate	C_22_H_18_O_10_	441.00	441.08 [44]	1.208	+
6	Caffeic acid	C_9_H_8_O_4_	179.00	179.03 [42]	1.378	+
7	Chlorogenic acid	C_16_H_18_O_9_	353.00	353.09 [42]	1.117	+
8	Arbutin	C_12_H1_6_O_7_	271.20	271.20 [45]	1.304	+
9	Trimethoxyflavone	C_18_H_16_O_5_	312.00	315.00 [46]	1.380	+

+++: high abundance, ++: abundant, +: low abundance.

**Table 2 pharmaceuticals-15-01156-t002:** Drug likeness and bioavailability score in in silico prediction of AHEE.

	Lipinski	Ghose	Veber	Egan	Bioavailability Score
**Gallic acid**	Yes	No (2 violations)	Yes	Yes	0.56
**Syringic acid**	Yes	Yes	Yes	Yes	0.56
** *trans* ** **-Ferulic acid**	Yes	Yes	Yes	Yes	0.85
**Myricetin**	Yes (1 violation)	Yes	No (1 violation)	No (1 violation)	0.55
**Catechin gallate**	Yes (1 violation)	Yes	No (1 violation)	No (1 violation)	0.55
**Caffeic acid**	Yes	Yes	Yes	Yes	0.56
**Chlorogenic acid**	Yes	No (1 violation)	No (1 violation)	No (1 violation)	0.11
**Arbutin**	Yes	No (1 violation)	Yes	Yes	0.55
**Trimethoxyflavone**	Yes	Yes	Yes	Yes	0.55

**Table 3 pharmaceuticals-15-01156-t003:** Absorption prediction of *A. halimus* leaf ethanolic extract.

	Water Solubility	Caco-2 Permeability	Intestinal Absorption	Skin Permeability	P-gp Substrate	P-gp I Inhibitor	P-gp II Inhibitor
Units	Log mol/L	Log Papp in 10^−6^ cm/s	%	cm/s	Categorical (Yes/No)		
**Gallic acid**	−1.64	−0.08	43.37	−2.73	No	No	No
**Syringic acid**	−1.84	0.49	73.07	−2.73	No	No	No
**Trans-ferulic acid**	−2.11	0.17	93.68	−2.72	No	No	No
**Myricetin**	−3.01	0.09	65.93	−2.73	No	No	No
**Catechin gallate**	−3.70	−1.26	62.09	−2.73	No	No	Yes
**Caffeic acid**	−1.89	0.63	69.40	−2.72	No	No	No
**Chlorogenic acid**	−1.62	−0.84	36.37	−2.73	No	No	No
**Arbutin**	−0.71	0.00	38.02	−2.80	No	No	No
**Trimethoxyflavone**	−4.11	1.40	98.07	−2.57	No	Yes	Yes

**Table 4 pharmaceuticals-15-01156-t004:** Distribution characteristics prediction of *A. halimus* leaf ethanolic extract. (**1**) Gallic acid, (**2**) syringic acid, (**3**) *trans*-ferulic acid, (**4**) myricetin, (**5**) catechin gallate, (**6**) caffeic acid, (**7**) chlorogenic acid, (**8**) arbutin, (**9**) trimethoxyflavone.

	1	2	3	4	5	6	7	8	9
**VDss (human)**	−1.85	−1.44	−1.36	1.31	0.66	−1.09	0.58	0.02	−0.29
**BBB permeability (Log BB)**	−1.10	−0.19	−0.23	−1.49	−1.84	−0.64	−1.40	−0.96	−0.20
**CNS permeability (Log PS)**	−3.74	−2.70	−2.61	−3.70	−3.74	−2.60	−3.85	−3.55	−2.14

**Table 5 pharmaceuticals-15-01156-t005:** Metabolism parameter prediction of *A. halimus* leaf ethanolic extract. (**1**) Gallic acid, (**2**) syringic acid, (**3**) trans-ferulic acid, (**4**) myricetin, (**5**) catechin gallate, (**6**) caffeic acid, (**7**) chlorogenic acid, (**8**) arbutin, (**9**) trimethoxyflavone.

	1	2	3	4	5	6	7	8	9
**CYP2D6 substrate**	No	No	No	No	No	No	No	No	No
**CYP3A4 substrate**	No	No	No	No	No	No	No	No	Yes
**CYP2D6 inhibitor**	No	No	No	No	No	No	No	No	Yes
**CYP3A4 inhibitor**	No	No	No	Yes	No	No	No	No	Yes

**Table 6 pharmaceuticals-15-01156-t006:** Excretion parameter prediction of AHEE. (**1**) Gallic acid, (**2**) syringic acid, (**3**) *trans*-ferulic acid, (**4**) myricetin, (**5**) catechin gallate, (**6**) caffeic acid, (**7**) chlorogenic acid, (**8**) arbutin, (**9**) trimethoxyflavone.

	1	2	3	4	5	6	7	8	9
**Total clearence (Log mL/min/Kg)**	0.51	0.64	0.62	0.42	−0.16	0.50	0.30	0.52	0.28
**Renal OCT2 substrate**	No	No	No	No	No	No	No	No	Yes

**Table 7 pharmaceuticals-15-01156-t007:** Toxicity characteristic prediction for *A. halimus* leaf ethanolic extract phytoconstituents. (**1**) Gallic acid, (**2**) syringic acid, (**3**) *trans*-ferulic acid, (**4**) myricetin, (**5**) catechin gallate, (**6**) caffeic acid, (**7**) chlorogenic acid, (**8**) arbutin, (**9**) trimethoxyflavone.

Molecules	1	2	3	4	5	6	7	8	9
**LD_50_ (mg/Kg)**	2000	1700	1772	159	1000	2980	5000	2500	5000
**Class**	4	4	4	3	4	5	5	5	5
**Hepatotoxicity**	Inactive	Inactive	Inactive	Inactive	Inactive	Inactive	Inactive	Inactive	Inactive
**Carcinogenicity**	Active	Inactive	Inactive	Active	Inactive	Active	Inactive	Inactive	Inactive
**Immunotoxicity**	Inactive	Inactive	Active	Inactive	Inactive	Inactive	Active	Inactive	Inactive
**Mutagenicity**	Inactive	Inactive	Inactive	Active	Inactive	Inactive	Inactive	Inactive	Inactive
**Cytotoxicity**	Inactive	Inactive	Inactive	Inactive	Inactive	Inactive	Inactive	Inactive	Inactive

**Table 8 pharmaceuticals-15-01156-t008:** Free radical scavenging and antioxidant capacity of AHEE. Values are expressed as mean ± SEM (*n* = 3).

Extract/Reference	DPPH Scavenging Capacity IC_50_ (mg/mL)	β-Carotene Bleaching Assay (mg/mL)	ABTS Scavenging (TE µmol/mL)	Iron Chelation	Total Antioxidant Capacity *
AHEE	0.36 ± 0.05	2.91 ± 0.14	44.10 ± 2.92	27.40 ± 1.46	124 ± 1.27
Ascorbic acid (AA)	0.19 ± 0.02	-	5.04 ± 0.78	0.94 ± 0.02	-
Butylated hydroxyanisole (BHA)	-	0.095 ± 0.00	-	-	-

* Total antioxidant capacity expressed as µg ascorbic acid equivalents/mg extract. TE: Trolox equivalent.

**Table 9 pharmaceuticals-15-01156-t009:** IC_50_ values and selectivity indexes of *A. halimus* ethanolic extract on MCF-7 and MDA-MB-231 human breast cancer cell lines.

Treatments	IC_50_ Value ± SD (µg/mL) *			Selectivity Index **	
	MCF-7	MDA-MB-231	PBMC	MCF-7	MDA-MB-231
**AHEE**	27.85 ± 3.14	51.95 ± 7.03	743.6 ± 9.55	26.7	14.31
**Cisplatin**	3.66 ± 1.05	1.60 ± 1.19	29.83 ± 1.19	8.15	18.64

* Values are obtained from three independent experiments and expressed as means ± SD. ** Selectivity index = (IC_50_ of PBMC/IC_50_ of tumor cells).

**Table 10 pharmaceuticals-15-01156-t010:** Target proteins of the identified compounds using SwissTarget Prediction. (1) Gallic acid, (2) syringic acid, (3) *trans*-ferulic acid, (4) myricetin, (5) catechin gallate, (6) caffeic acid, (7) chlorogenic acid, (8) arbutin, (9) methoxyflavone.

PDB IDs	Protein Target Name	Resolution	Native Ligands	Ligands								
				1	2	3	4	5	6	7	8	9
**12CA**	Human carbonic anhydrase II	2.40 Å	Tetrazole [123]	**X**	**X**	**X**			**X**	**X**		
**3MDZ**	Human carbonic anhydrase VII	2.32 Å	Acetazolamide [124]	**X**	**X**	**X**			**X**	**X**	**X**	
**1AZM**	Human carbonic anhydrase I	2.00 Å	Furosemide [125]	**X**	**X**	**X**	**X**	**X**	**X**	**X**		
**2CKJ**	Human milk xanthine oxidoreductase	3.59 Å	Oxypurinol [126]				**X**					
**1ZXM**	Human Topo IIa ATPase/AMP-PNP	1.87 Å	1,2-Benzenedicarboxylic acid, mono(2-ethylhexyl) ester [127]				**X**					
**2JKV**	Human phosphogluconate dehydrogenase	2.53 Å	NADPH [128]					**X**				
**1G5M**	Human antiapoptotic protein BCL-2	1.80 Å	ABT-737 [129]					**X**				
**3IAR**	Human adenosine deaminase	1.52 Å	2′-Deoxyadenosine [130]								**X**	
**1M73**	Human purine nucleoside phosphorylase (PNP)	2.30 Å	Forodesine [131]								**X**	
**5N2S**	Human A_1_ adenosine receptor	3.30 Å	PSB36 [132]									**X**
**2YDO**	Human A_2A_ adenosine receptor	3.00 Å	Istradefylline [132]									**X**
**5NJ3**	ABC transporter	3.78 Å	Gefitinib [133]									**X**

## Data Availability

Data is contained within the article and Appendix A.

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
