# Peer review of "LC–MS/MS Phytochemical Profiling, Antioxidant Activity, and Cytotoxicity of the Ethanolic Extract of Atriplex halimus L. against Breast Cancer Cell Lines: Computational Studies and Experimental Validation"

_pharmaceuticals, 2022, doi:10.3390/ph15091156_

Round 1
Reviewer 1 Report
The manuscript is generally well structured and well written. In addition, the topic currently addressed is of great importance and any advance, no matter how small, is important in this type of disease.
In the attached file I indicate comments that should be addressed by the authors of the manuscript.

Author Response
Response to Reviewers
|
Manuscript title: |
LC-MS/MS Phytochemical Profiling, Antioxidant Activity, and Cytotoxicity of the Ethanolic Extract of Atriplex halimus L. Against Breast Cancer Cell Lines: Computational studies and Experimental Validation |
|
Manuscript ID: |
Pharmaceuticals-1894527 |
Dear Editors, Dear Reviewers,
Thank you for giving us the opportunity to improve our manuscript with the revised version and thank you for your useful comments.
We really thank the Reviewers for their thorough review. We hope this revision will satisfy reviewers’ queries, and that our work will be considered for publication in Pharmaceuticals. Below we have made efforts to either address or respond to each (paraphrased) requested change and communicated weakness. Major changes are highlighted in yellow text in the revision. All typos/minor concerns have been fixed in the manuscript and are otherwise not addressed in this response.
With kind regards
Dr. Hano, Dr. Addi, and the co-Authors
Reviewer 1. The manuscript is generally well structured and well written. In addition, the topic currently addressed is of great importance and any advance, no matter how small, is important in this type of disease. In the attached file I indicate comments that should be addressed by the authors of the manuscript.
Responses to R1. thank you very much for your kind comments. We have paid particular attention to all your comments and suggestions, and provided you an answer to each below as well as in the revised version of the manuscript appearing in yellow. We hope our answers will clarify our point and improve the quality of this work in order to make it suitable for publication in Pharmaceuticals.
Response (Line 122). We have added references that support our statement as suggested. Thank you.
Line 127. What could be the reason of the higher availability of trans-ferulic acid?
Response. The Bioavailability score (BAS) aims to forecast the likelihood that a substance will have Caco-2 permeability that can be measured or at least 10% oral bioavailability in rats. BAS is 0.11 for anions for which PSA is >150 Å2, 0.56 if PSA is between 75 and 150 Å2, and 0.85 if PSA is < 75 Å2. For the remaining compounds BAS is 0.55 if it passes the rule-of-five (Lipinski’s rule of five) and 0.17 if it fails. In our case, the observed BAS for trans-ferulic acid can be justified by the fact that the later compound has a topological polar surface area (PSA) < 75 Å2 (TPSA (trans-ferulic acid) = 66.8 â„«²), thus, the bioavailability score is 0.85.
We have added a paragraph about trans-ferulic acid in our paper; Thank you so much for your remark
Comment. Would it be possible to compare these cytotoxicity results with other extracts analyzed by other authors?
Response.
LC-MS/MS Profiling of AHEE comment. We have translated your pertinent comment and the response is below.
Response. We thank the reviewer for his remarks. The LC-MS/MS procedure followed in this paper is already published by our team as well as by other authors. The following papers used the same methodology in their profiling (e.g. the following three papers).
- https://doi.org/10.3390/molecules25215073
- https://doi.org/10.3390/molecules26237300
- https://doi.org/10.3390/nu13082759
We have added a reference for the used methodology in the manuscript. Thank you again.
Line 450. Why was the extraction performed with ethanol instead of another green solvent such as water, which is cheaper and more environmentally friendly?
Response. The main reason why we have used ethanol, is that the later can dissolve both polar and non-polar metabolites found in the plant material. Ethanol is the second most important solvent after water. Ethanol is the least toxic of the alcohols (only poisonous in large amounts), which makes it more suitable for use for such experiments.
Line 516. Where these two cells were used for a specific reason? Could different ones be studied?
Response. Female breast cancer has distinctive histological and biological characteristics, clinical presentations, and therapeutic responses. Triple-negative breast cancer (TNBC) is the most invasive subtype [1–3]. TNBC is characterized by insensitivity to targeted therapies since it has a lack of progesterone receptor (PR-), and estrogen receptor (ER-) expression as well as HER2 (human epidermal growth factor receptor 2) amplification. MDA-MB-231 is a highly aggressive and invasive TNBC cell line, and MCF-7 is one of few breast cancers cell line that express the estrogen receptor (ER+). Considering that Atriplex halimus has been used in phytotherapy against breast cancer for a long time, the in vitro analysis in this study was performed on these breast cancer cell lines to determine if Atriplex halimus extract had an antiproliferative effect, and to determine whether the activity was estrogen-dependent.
More cancer cell lines that represent the other subtypes of breast cancer, such as BT474, MDA-MB-468, MDA-MB-453, and SKBR3, can be used to develop further conclusions and gain insight into the effect of AHEE on breast cancer.
Bibliography:
- Testa, U.; Castelli, G.; Pelosi, E. Breast Cancer: A Molecularly Heterogenous Disease Needing Subtype-Specific Treatments. Med. Sci. 2020, 8, 18.
- Barzaman, K.; Karami, J.; Zarei, Z.; Hosseinzadeh, A.; Kazemi, M.H.; Moradi-Kalbolandi, S.; Safari, E.; Farahmand, L. Breast Cancer: Biology, Biomarkers, and Treatments. Int. Immunopharmacol. 2020, 84, 106535.
- Yersal, O.; Barutca, S. Biological Subtypes of Breast Cancer: Prognostic and Therapeutic Implications. World J. Clin. Oncol. 2014, 5, 412.
Reviewer 2 Report
The manuscript deals with LC-MS/MS profiling of ethanolic extract from Atriplex halimus and in vitro antioxidant and cytotoxic activity on some breast cancer cell lines. The authors correlated the experimental data from in vitro test with in silico ADME and molecular docking calculations on molecules identified in LC-MS/MS profiling.
There are a few recommendations:
1. Please, add TIC or base peak chromatograms of the extract.
2. In Table 1, please add product ions of each identified compound's MS/MS spectra.
3. In table 1 again, please specify the ion mode (negative or positive). Usually, in the positive mode, you detect protonated molecules [M+H]+, while in the negative, you detect deprotonated molecules [M-H]-.
4. In the experimental section, please specify the mode of detection (positive or negative) in LC-MS/MS analysis.
Author Response
Response to Reviewers
|
Manuscript title: |
LC-MS/MS Phytochemical Profiling, Antioxidant Activity, and Cytotoxicity of the Ethanolic Extract of Atriplex halimus L. Against Breast Cancer Cell Lines: Computational studies and Experimental Validation |
|
Manuscript ID: |
Pharmaceuticals-1894527 |
Dear Editors, Dear Reviewers,
Thank you for giving us the opportunity to improve our manuscript with the revised version and thank you for your useful comments.
We really thank the Reviewers for their thorough review. We hope this revision will satisfy reviewers’ queries, and that our work will be considered for publication in Pharmaceuticals. Below we have made efforts to either address or respond to each (paraphrased) requested change and communicated weakness. Major changes are highlighted in yellow text in the revision. All typos/minor concerns have been fixed in the manuscript and are otherwise not addressed in this response.
With kind regards
Dr. Hano, Dr. Addi, and the co-Authors
Reviewer 2. The manuscript deals with LC-MS/MS profiling of ethanolic extract from Atriplex halimus and in vitro antioxidant and cytotoxic activity on some breast cancer cell lines. The authors correlated the experimental data from in vitro test with in silico ADME and molecular docking calculations on molecules identified in LC-MS/MS profiling. There are a few recommendations:
Response to R2. Dear Reviewer 2, we highly appreciate your comments on our paper, and we thank you for pointing errors in our manuscript. We have added the necessary changes to the manuscript, and we hope the revised file satisfies your queries.
Recommendation 1. Please, add TIC or base peak chromatograms of the extract.
Response. Dear reviewer, since the identification method used by the Shimadzu Ultra High-Performance Liquid Chromatograph (UHPLC; Nexera XR LC 40), is based on the identification of the typical fragments of the molecules using a double mass spectrometer detector. there is no actual separation of the molecules in the column like other HPLC machine. If we generate a base peak, it will be similar to that below. All data of the analysis are in Table 1 and typical fragments which all can be found in the supplementary material.
Recommendation 2. In Table 1, please add product ions of each identified compound's MS/MS spectra.
Response. We thank the reviewer for his/her remark, for this analysis the method for the identification using this machine was developed in-house including the database using standard compounds for the molecules. The identification database relies only on the typical fragment characterized for each molecule. The MS/MS was operated in negative electrospray ionization (ESI-) setting the nebulizing gas flow at 3 L/min, heating gas flow 10 L/min, interface temperature 370 °C, DL temperature 250 °C, heat block temperature 450 °C, drying gas flow 10 L/min.
Recommendation 3. In table 1 again, please specify the ion mode (negative or positive). Usually, in the positive mode, you detect protonated molecules [M+H]+, while in the negative, you detect deprotonated molecules [M-H]-.
Response. We highly appreciate your pertinent comment. As stated in the spectra enclosed to the supplementary material, the utilized ionization mode was ESI-. Table 1 was corrected as well according to the given indications and the [M-H]- values from the literature.
Recommendation 4. In the experimental section, please specify the mode of detection (positive or negative) in LC-MS/MS analysis.
Response. We really appreciate your remarks and we accept the revision. We have specified the mode of detection (which is the negative mode; in negative electrospray ionization ESI-) in the paper.

Reviewer 3 Report
This paper investigates the chemical composition, antioxidant activity and cytotoxicity against breast cancer cell lines of the ethanolic extract of Atriplex halimus L. leaves (AHEE). Although this paper provides a lot of data, the evidence provided in the chemical composition analysis is not enough to support the correctness of the nine chemical compositions obtained, resulting in many subsequent computations and predictions based on these nine compositions may be wrong. Therefore, I do not recommend that this article be published in this journal, and suggest that the authors reconfirm the chemical composition analysis. Here are some suggestions:
1. Introduction: The review of the previous research literature on A. halimus L. is insufficient and too brief, especially the introduction of the content related to this paper should be more detailed.
2. Section 2.1: It is unscientific and dangerous for the authors to identify the 9 main components of AHEE (Table 1, Supplementary file 1 Figure 1 and Supplementary file 2) by only LC-MS/MS analysis, and many of the data in the paper are based on computations and predictions for these nine compounds. The retention time interval of these nine components on the LC spectrum was only as short as in 1.3 min, and there is no other literature to support this inference. It is difficult to judge the structure of the compounds in this way, and it is recommended to provide more analytical data to support the authors' inference.
3. Figure 1: Please check if the chemical structure of caffeic acid is correct.
4. Line 128: It should be “nine compounds”.
5. Section 2.5.1: The antioxidant activity should be compared with other literature, such as Bouaziz et al. (2021). Exp Parasitol. 229:108155.
6. Figure 5 x-axis: Check the word “Concentration”.
7. Table 8: The units are different from those shown in Abstract section.
8. Notes below Table 9: If it should be “selectivity index = IC50 of PBMC/IC50 of tumor cells”?
9. The format of the Reference section needs to be checked in detail. There are differences between the format of the reference section of this article and the format prescribed by this journal, please revise carefully.
Author Response
Response to Reviewers
|
Manuscript title: |
LC-MS/MS Phytochemical Profiling, Antioxidant Activity, and Cytotoxicity of the Ethanolic Extract of Atriplex halimus L. Against Breast Cancer Cell Lines: Computational studies and Experimental Validation |
|
Manuscript ID: |
Pharmaceuticals-1894527 |
Dear Editors, Dear Reviewers,
Thank you for giving us the opportunity to improve our manuscript with the revised version and thank you for your useful comments.
We really thank the Reviewers for their thorough review. We hope this revision will satisfy reviewers’ queries, and that our work will be considered for publication in Pharmaceuticals. Below we have made efforts to either address or respond to each (paraphrased) requested change and communicated weakness. Major changes are highlighted in yellow text in the revision. All typos/minor concerns have been fixed in the manuscript and are otherwise not addressed in this response.
With kind regards
Dr. Hano, Dr. Addi, and the co-Authors
Reviewer 3. This paper investigates the chemical composition, antioxidant activity and cytotoxicity against breast cancer cell lines of the ethanolic extract of Atriplex halimus L. leaves (AHEE). Although this paper provides a lot of data, the evidence provided in the chemical composition analysis is not enough to support the correctness of the nine chemical compositions obtained, resulting in many subsequent computations and predictions based on these nine compositions may be wrong. Therefore, I do not recommend that this article be published in this journal, and suggest that the authors reconfirm the chemical composition analysis. Here are some suggestions:
Response to R3. We would like to express our appreciation for your careful reading of this work and your insightful comments that encourage and help us to improve our article. Therefore, according to the suggestions and comments, we have revised thoroughly our manuscript and the responses to the comments are listed below.
Suggestion 1. Introduction: The review of the previous research literature on A. halimus L. is insufficient and too brief, especially the introduction of the content related to this paper should be more detailed.
Response 1. Dear Reviewer thank you for your sharp remark, we have added more details about the studied plant (Atriplex halimus L.).
Suggestion 2. Section 2.1: It is unscientific and dangerous for the authors to identify the 9 main components of AHEE (Table 1, Supplementary file 1 Figure 1 and Supplementary file 2) by only LC-MS/MS analysis, and many of the data in the paper are based on computations and predictions for these nine compounds. The retention time interval of these nine components on the LC spectrum was only as short as in 1.3 min, and there is no other literature to support this inference. It is difficult to judge the structure of the compounds in this way, and it is recommended to provide more analytical data to support the authors' inference.
Response 2. Dear reviewer, we understand your concern about the LC-MS/MS analysis for that let us clarify how it is done. We actually consider the analysis one of the strength points of this study. The extract analysis was done using a Shimadzu Ultra High-Performance Liquid Chromatograph (UHPLC; Nexera XR LC 40), coupled to a MS/MS detector, which is considered one of the most powerful and performant UHPLC machines in the market. The identification was very accurate using a recently updated library. The analysis was performed with in flow injection (FIA= flow injection analysis). This means that there is not a chromatographic separation because the analysis is not based on retention time and separation. the identification with this Shimadzu machine is done according to the molecular weight of the typical fragments of molecules after the fragmentation of the molecules. two performant Mass spectrometers are able perfectly to identify the fragments and assign them to the corresponding molecule.
There are several papers published using this same machine (and the same methodology) e.g.
- https://doi.org/10.3390/nu13082759
- https://doi.org/10.3390/toxins14070475
Suggestion 3. Figure 1: Please check if the chemical structure of caffeic acid is correct.
Response 3. Thank you very much for your remark. We have rectified the double bond in the chemical structure of Caffeic acid.
Suggestion 4. Line 128: It should be “nine compounds”.
Response 4. We have rectified the number of compounds. Thank you so much.
Suggestion 5. Section 2.5.1: The antioxidant activity should be compared with other literature, such as Bouaziz et al. (2021). Exp Parasitol. 229:108155.
Response 5. We thank the reviewer for his comment. We have added additional parts of the text to the revised manuscript.
Suggestion 6. Figure 5 x-axis: Check the word “Concentration”.
Response 6. Thank you for you remark we have made the necessary correction.
Suggestion 7. Table 8: The units are different from those shown in Abstract section.
Response 7. We thank the reviewer for his remark, we have made the necessary change in the abstract section.
Suggestion 8. Notes below Table 9: If it should be “selectivity index = IC50 of PBMC/IC50 of tumor cells”?
Response 8. Thank you for your pertinent remark, we have rectified it in the manuscript.
Suggestion 9. The format of the Reference section needs to be checked in detail. There are differences between the format of the reference section of this article and the format prescribed by this journal, please revise carefully.
Response 9. We have checked the reference style used and we have change it to Pharmaceuticals’ referencing style. Thank you for your remark.
Round 2
Reviewer 3 Report
The authors have already responded and revised related issues, which are already acceptable, so it is recommended to publish.